# What Is the Value of DXA in Predicting Fracture Risk in Postmenopausal Women? A 10-Year Follow-Up Study in the Małopolska Region

**DOI:** 10.3390/biomedicines13122955

**Published:** 2025-12-01

**Authors:** Przemysław Borowy, Bogdan Batko, Alicja Kamińska, Patrycja Major, Katarzyna Gołojuch, Jakub Smyk, Krzysztof Batko, Edward Czerwiński

**Affiliations:** 1Department of Rheumatology and Immunology, J. Dietl Hospital, 31-121 Krakow, Poland; 2Faculty of Medicine and Health Sciences, Andrzej Frycz Modrzewski, Krakow University, 30-705 Krakow, Poland; 3Stefan Żeromski Specialist Hospital, 31-913 Krakow, Poland; 45th Military Clinical Hospital in Kraków, 30-901 Krakow, Poland; 5Ludwik Rydygier Memorial Hospital, 31-820 Krakow, Poland; 6Department of Dermatology and Allergology, University Hospital, 30-688 Krakow, Poland; 7Doctoral School of Medical and Health Sciences, Jagiellonian University, 31-202 Krakow, Poland; 8Krakow Medical Center, 31-501 Krakow, Poland

**Keywords:** fracture risk, DXA, BMD, T-score, polish population, fracture prediction

## Abstract

**Background:** Bone mineral density (BMD) assessed by DXA is a well-established predictor of osteoporotic fracture risk. However, data regarding the Polish female population remains limited. **Objective:** To evaluate the predictive value of BMD measurements for vertebral, hip, and all low-energy fractures in women aged 50 years and older. **Methods:** A total of 1.311 women from the Małopolska region underwent BMD assessment at the femoral neck, lumbar spine. The average follow-up period was 10.2 years, during which 479 osteoporotic fractures were recorded. **Results:** DXA measurements at the femoral neck showed the strongest correlation with hip fracture risk. Each one standard deviation decrease in the femoral neck T-score increased the risk of hip fracture by 2.1 times (HR 2.10; 95% CI 1.28–3.46; *p* = 0.003), after adjusting for age, but is not linear. A 1 SD decrease in the hip T-score was associated with a 28% increase in the risk of all osteoporotic fractures (HR 1.28; CI 1.17–1.40; *p* < 0.001), 53% increase in vertebral (HR 1.53; CI 1.13–2.08; *p* = 0.006) and 30% in hip (HR 1.30; CI 0.81–2.09; *p* = 0.278). The AUC values for hip BMD and hip T-score had the highest predictive value—AUC (area under the curve was 0.732 and 0.720, *p* < 0.01). **Conclusions:** BMD at the femoral neck proved to be a stronger predictor of hip fractures than measurements at the spine, radius. The risk increase associated with BMD/T-score reduction was non-linear. These findings confirm results from other benchmark studies.

## 1. Introduction

Osteoporotic fractures are the most severe consequence of osteoporosis, causing disability, mortality, chronic pain, and substantial social and economic burdens; they remain the strongest predictor for subsequent fractures. The lifetime risk of sustaining an osteoporotic fracture is estimated at 46.4% in women and 22.4% in men [1], exceeding that of stroke and breast cancer [2]. The most significant risk factor for osteoporotic fracture is low bone mineral density (BMD). Assessment by dual-energy X-ray absorptiometry (DXA), as recommended by the World Health Organization (WHO), constitutes the gold standard for diagnosis and the primary determinant for clinical management [3,4]. The prognostic utility of DXA in fracture risk prediction has been extensively evaluated in multiple large-scale studies worldwide [5], although only one such investigation has included the postmenopausal female population in Poland [6]. Since 2009, no prospective data on osteoporotic fractures have been published in Poland. It is well established that even within the European population of Caucasian women, substantial differences in the incidence of low-energy fractures are observed [7]. These differences are evident even between various regions within the same country [8]. The objective of the present study was to evaluate the predictive value of lumbar spine and femoral neck DXA measurements in estimating the risk of vertebral fractures, hip fractures, and all low-energy fractures in a population-based, randomly selected cohort of postmenopausal women from the Małopolska region, during a 10-year prospective follow-up of a large cohort. We analyzed BMD, T-scores, and Z-scores in each patient.

The presentation of current epidemiological data, even when limited to a local population, will enable more accurate future estimates of fracture incidence in an aging society. Such data may support the selection of screening and therapeutic programs funded by the National Health Fund and facilitate projections of future fracture-related healthcare costs. Moreover, it may be used to refine the FRAX calculator and determine the optimal treatment threshold for the Polish population.

## 2. Materials and Methods

A total of 5.092 randomly selected Caucasian women, aged 50–79 years, who were patients of the Osteoporosis Outpatient Clinic, were invited to participate in the study. Between 1997 and 2003, these individuals underwent densitometric assessment of the lumbar spine and/or femoral neck. Complete medical documentation, including a fracture risk assessment questionnaire, records from outpatient and inpatient treatment, as well as imaging results (X-ray, CT, MRI confirming fractures and comorbidities), was obtained for 3.350 participants (Figure 1). The questionnaire included items on demographic and anthropometric data (such as maximum lifetime height), past and comorbid diseases, and risk factors incorporated into the FRAX algorithm. A basic gynecological history was obtained from each woman (age at menarche and menopause, number of pregnancies, use of contraception, and hormone therapy). All participants were asked about a diagnosis of osteoporosis and its treatment. Treatment was defined as the use of one of the active medications for at least 12 months. Data on fractures were verified based on the provided medical documentation. Fractures that occurred before the age of 50 were not analyzed in this study.

Lumbar spine and hip DXA examinations were performed in accordance with the ISCD methodology [9], using a Hologic Delphi densitometer (Hologic Inc., Marlborough, MA, USA, software version 11.1L) in all patients. T-scores for the femoral neck were calculated using the NHANES III reference database, whereas lumbar spine T-scores were derived from the Hologic manufacturer’s reference population. After a mean follow-up period of approximately 10 years, a structured telephone interview was conducted using a questionnaire analogous to that employed at baseline, supplemented with all available medical records.

For 34% of participants, we had access to questionnaires and complete medical documentation from subsequent years, as these patients were regularly monitored at our center. The collected information was then reviewed by physicians experienced in the management of osteoporosis. Individuals with incomplete medical documentation (*n* = 2016), those lost to follow-up (*n* = 1.663), those who declined participation in the second survey (*n* = 210), or those who had died (*n* = 143) were excluded from the study. Fractures were assessed in two ways. In patients regularly followed at the osteoporosis clinic, spine X-rays and/or vertebral fracture assessment (VFA) were performed in cases of back pain, a height loss greater than 4 cm, or body deformity detected on physical examination. In patients not receiving regular follow-up, fractures were confirmed based on the provided medical documentation.

Ultimately, 1.311 women with complete and reliable medical data were included in the final analysis (Table 1). The analysis focused on the incidence of major fractures, defined as fractures of the spine, distal radius, distal humerus, and proximal femur.

The study received a positive opinion from the Bioethics Committee (Ref. No. OIL/KBL/381/2025, approved on 30 September 2025).

### Statistical Analysis

We used R version 4.4.3 (R Core Team, 2025; R Foundation for Statistical Computing, Vienna, Austria) with the use of tidyverse, survival, survminer, and pROC packages.

Risk was modeled using Cox proportional hazard regression to estimate hazard ratios (HRs) and 95% confidence intervals (CIs) to model the associations between bone density assessments and fracture risk. Linear models assessed risk per 0.1 g/cm^2^ decrease in BMD and per standard deviation decrease in T-scores and Z-scores. Categorical analyses used clinically pragmatic thresholds to examine potential non-linear relationships. The low event rate necessitates caution in interpretation and further validation. Receiver operating characteristic (ROC) curves evaluated discriminative performance, with area under the curve (AUC) and optimal cutpoints calculated using Youden’s index. Complete case analysis was performed on a model construct level, with sample sizes varying by DXA parameter availability. Statistical significance was defined as two-sided *p* < 0.05.

## 3. Results

### 3.1. Study Population

During a mean (SD) follow-up of 10.2 (1.6) years (range 5–18), 502 incident fractures were recorded among 1311 postmenopausal women. Overall, osteoporotic fracture incidence was 38.3%. These included 61 vertebral, 22 femoral neck, and 292 distal radius fractures. Complete BMD data were available for 1214 (92.6%) and 870 (66.4%) study participants at the hip and lumbar spine, respectively.

### 3.2. Site-Specific Risk Estimates

We observed that a decline in bone density measures (BMD, T-score, and Z-scores) at both the lumbar spine and proximal femur was significantly associated with increased fracture risk across different fracture locations (Table 2).

Hip parameters showed the strongest and most consistent associations, particularly for femoral neck and composite osteoporotic fractures. By comparison, lumbar spine BMD showed more modest associations for all-site and vertebral fracture risk, but no significant association with hip fractures.

The relationship between bone density and fracture risk may be non-linear, as implied by categorical analyses with variable risk across BMD and T-score strata. Models by category are presented below (Table 3, Table 4 and Table 5 for vertebral, hip, and all-site fractures, respectively).

#### 3.2.1. Risk of Vertebral Fracture

Categorical analyses suggest a progressive increase in vertebral fracture risk with lower BMD and T-score categories (Table 3). For lumbar spine BMD, participants with values below 0.75 g/cm^2^ demonstrated a twofold increased risk compared with those having BMD ≥ 0.95 g/cm^2^. Similarly, the lowest femoral neck BMD category (<0.65 g/cm^2^) was associated with over two-fold increase in vertebral fracture risk relative to the reference group. T-score categorization was characterized by even stronger risk associations.

#### 3.2.2. Risk of Hip Fracture 

Femoral neck parameters were linked with the strongest association of hip fracture, which remains consistent with the site-specific nature of bone loss (Table 4). Categorical analyses suggest a threshold effect, with higher risk observed mainly in the lowest hip BMD and T-score strata. In contrast, lumbar spine measures showed weaker relationships. The wide CIs reflect sample size constraints and warrant caution until further validation.

#### 3.2.3. Risk of Osteoporotic Fracture (All Sites)

A dose–response relationship was observed with declining BMD and T-scores, particularly for parameters of the femur neck. Hip-based assessments were characterized by the best discrimination, while lumbar spine-derived assessments were weaker.

### 3.3. Receiver Operating Characteristic Analysis

We performed receiver operating characteristic (ROC) analyses to quantify the discriminative performance of models based on DXA parameters at the hip and vertebral level. This approach is time-agnostic, which leads to a certain degree of bias in terms of variable exposure time frame.

A comparative visualization of ROCs for different sites is shown in Figure 2.

#### 3.3.1. Risk of Vertebral Fracture

For spinal fractures, AUC values ranged between 0.593 and 0.639 across different DXA assessments (Table 6), which implies weak performance. Lumbar spine BMD was characterized by the highest discriminatory ability, though the corresponding sensitivity and specificity (given low outcome prevalence) preclude clinical utility. Lumbar spine T scores showed similarly weak results. Of note, hip-based measures showed lower, but comparable AUC values for prediction of spinal fractures (0.593–0.597).

#### 3.3.2. Risk of Hip Fracture

Hip DXA parameters showed markedly better discriminative ability for femoral neck fractures, as compared with spinal measurements (Table 7). Femoral neck BMD was characterized with the best performance at the 0.74 g/cm^2^ cut-off, with sensitivity and specificity satisfactory for adjunct clinical use. Hip T scores performed similarly. By contrast, spinal measurements showed poor to modest discrimination potential.

#### 3.3.3. Risk of All-Site Fracture

When considering fractures at different sites, discrimination ability was limited (AUCs ranging from 0.560 to 0.621; Table 8). Femoral neck BMD and T-scores showed the highest discriminatory performance, but sensitivity and specificity did not exceed 65% at the optimal cut-offs.

## 4. Discussion

The occurrence of a fracture is multifactorial in origin but primarily results from compromised bone tissue strength. Therefore, osteoporosis diagnosis and fracture risk assessment are based on the measurement of BMD. In vivo studies have demonstrated an excellent correlation between mechanical strength, as assessed by femoral neck shear strength testing, and femoral neck BMD [10]. In a study of 26 participants, a correlation coefficient of 0.92 was reported between the maximum force applied before fracture and the BMD of the femoral trochanter [11]. Ex vivo investigations have shown that BMD assessment predicts femoral neck fracture risk with an accuracy of 65–80% when compared to direct measures of bone strength [12]. Similar studies have reported DXA prediction accuracy for vertebral fractures in the range of 73–79% [13]. The prognostic value of BMD and T-score, however, is relatively low when predicting the overall fracture risk, defined as the total of all osteoporotic fractures, as confirmed in our cohort [14]. DXA is particularly valuable for predicting vertebral and femoral neck fractures. The calculated fracture risk in our study was comparable to that reported in other publications. It was derived from the analysis of a large cohort and long-term follow-up of a Polish population of postmenopausal Caucasian women [15]. The results were consistent despite several important study limitations. The major limitation is the accurate assessment of fractures, particularly vertebral fractures, which are asymptomatic in most cases. During follow-up, active screening was performed in only 34% of patients, while the remaining participants were assessed solely based on medical history and provided documentation. However, reliance on retrospective data may introduce recall bias. This may have contributed to an underestimation of the true number of vertebral fractures. This explanation appears to us the most plausible.

A low number of hip fractures was also observed, which, among other factors, resulted in wide confidence intervals. This may reflect a specific characteristic of the Polish population; however, these findings should be interpreted with caution and verified using larger datasets.

The analysis included information on treatment use, but a precise evaluation of the effect of therapy on fracture risk was not feasible. At baseline, 6.2% of participants (79 women) received treatment according to the criteria, increasing to 16.9% (217 patients) after 10 years. This represents a relatively small proportion, despite a diagnosis of osteoporosis in 43.9% of the study population. Indirectly, this also highlights the problem of the lack of treatment despite an established diagnosis, a phenomenon widely described in the literature. Pharmacotherapy affects both fracture incidence and the predictive performance of DXA, particularly in patients with the lowest BMD values.

Due to statistical power concerns (e.g., highly variable event rate per variable), our primary analyses are presented based on a crude Cox model with no adjustment for potential confounders. In sensitivity analyses, we observed that the addition of age as a covariate led to modest attenuation of hazard ratios (<10%), with bone density measures consistently showing strong predictive capacity. We did not account for other clinically relevant covariates (e.g., smoking, medication use), which represents an important limitation of this report, as we are unable to account for this residual confounding.

In the first landmark publication of this type, a decrease in BMD of 0.1 g/cm^2^ was associated with an increased risk of femoral neck fracture (RR 1.9; CI, 1.4–2.7) [16]. In another study, the same decrease was associated with a different femoral neck fracture risk (RR 2.4; CI, 2.1–2.7) [5], likely due to differences in study populations. In our study, the same decrease was associated with an increased risk of RR 1.8. In a meta-analysis of 11 studies including a total follow-up of approximately 90.000 person-years, investigators demonstrated that fracture risk varies depending on the BMD measurement site. For spine BMD, the risk of all fractures was RR 1.5 (CI, 1.4–1.6), the risk of vertebral fractures was RR 2.3 (CI, 1.9–2.8), but for femoral neck fractures based on hip DXA, it was RR 2.6 (CI, 2.0–3.5) [5].

Our findings confirmed that all analyzed femoral neck DXA parameters (hip BMD, T-score, and Z-score) correlate most strongly with hip fracture risk. An abnormal hip T-score also represented the strongest risk factor for femoral neck fracture. The Cox model confirmed that any significant reduction in bone mass translates into an increased fracture risk. T-score analysis yielded similar results [17,18]. The RR values according to the measurement site are presented in Table 9.

As summarized in Table 9, the relative hip risks observed in our cohort are consistent with those reported in other large epidemiological studies, such as the SOF and the meta-analysis by Marshall et al. This confirms that the predictive value of DXA for hip fractures in Polish postmenopausal women is comparable to that observed in other Caucasian populations. The consistency of the results is partly attributable to the main strengths of the study: a large, well-defined cohort with a long follow-up period of approximately 10 years. Lower relative risk values than those used for comparison were observed for vertebral fractures, which may be attributable to the study limitations described above. In large-scale cohorts, such as the Manitoba BMD Registry, the number of reported fractures is likewise high, highlighting the advantage of such registries.

Lumbar spine densitometry results showed a slightly larger RR for spine fracture (RR 1.55) compared to hip fracture (RR 1.30). However, the sensitivity and specificity of these measurements were low, with the highest values observed for the spine T-score (sensitivity 67.3%, specificity 54.9%) and the femoral neck T-score (sensitivity 53.6%, specificity 68.7%). ROC curve analysis confirmed that hip densitometry (BMD hip) was the most accurate predictor of hip fractures, with a sensitivity of 80% and a specificity of 61.1%. This association was evident despite the relatively low number of hip fractures (37 in the entire cohort). AUC values below 0.7 for other measurement sites and fracture types suggest that an abnormal DXA result constitutes only one of several risk factors for fracture. This finding aligns with the current diagnostic approach in osteoporosis, which is based on identifying multiple modifiable and non-modifiable fracture risk factors [21], upon which therapeutic decisions are subsequently made.

The relationship between BMD, T-score, Z-score, and fracture risk was non-linear, with a sharp increase in RR observed at very low BMD values (lumbar spine < 0.75) or T-score spine < −4.0—threshold effect). The non-linear increase in risk with decreasing T-score is often overlooked when interpreting DXA results. However, it has important practical implications, as it indirectly affects treatment selection according to the criteria for very high, high, and low fracture risk. The differences in fracture risk observed in our study are reflected in current therapeutic recommendations (e.g., the use of anabolic agents in patients at very high fracture risk). The conclusions drawn from these data may support pharmacoeconomic analyses of osteoporosis treatments used in Poland and inform updates to reimbursement criteria for anabolic therapies in selected patient groups.

The results confirmed the well-established observation that fracture risk was highest for fractures occurring at the anatomical site where DXA was performed, namely, vertebral fractures for spine DXA and femoral neck fractures for hip DXA.

Although the study cohort consisted exclusively of Caucasian postmenopausal women from a single region (Małopolska, Poland), we believe that the results may be representative of the entire Polish population. According to statistical data, 98–99% of Polish women aged 50 years and older are of Caucasian origin, and the socioeconomic situation is comparable across different regions of the country [22]. Potential differences between urban and rural populations were not analyzed in this study.

## 5. Conclusions

Fracture risk increases non-linearly with decreasing BMD and T-score. This is most evident for hip DXA. The RR values for the Małopolska population were comparable to those reported in other published population-based studies; therefore, the WHO criteria appear to be well-suited to the studied population. The moderate sensitivity and specificity of DXA parameters indicate that DXA should be considered only one of several tools for fracture risk assessment. Clinical decision-making should also incorporate other risk factors, for example, by using the FRAX, FRAX plus, or other fracture risk calculators. The proven high risk of hip fracture in individuals with very low BMD should, however, be taken into account when developing therapeutic strategies for this patient group. Access to anabolic therapies should be readily available and reimbursed for these patients, which, at present, is very limited in Poland.

## Figures and Tables

**Figure 1 biomedicines-13-02955-f001:**
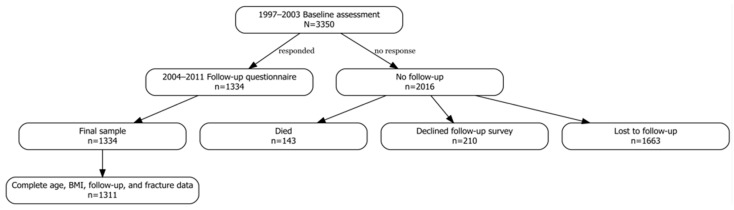
Flowchart illustrating participant selection for this study.

**Figure 2 biomedicines-13-02955-f002:**
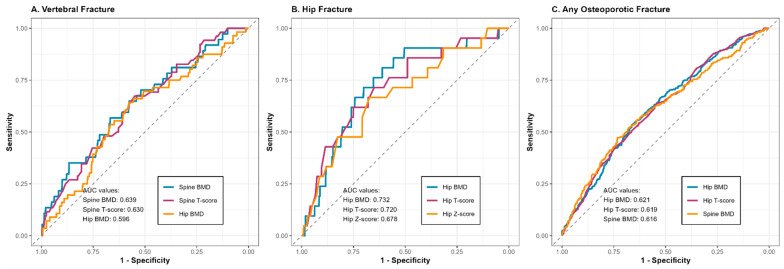
Plot for linear risk associations across measurements and fracture type.

**Table 1 biomedicines-13-02955-t001:** Baseline demographic characteristics of the participants (*N pts* = 1311).

Variable	Mean (SD)	Median (IQR)	Range
Age (years)	64.7 (7.1)	65 (60–70)	20–92
Height (m)	1.59 (0.06)	1.60 (1.56–1.64)	1.38–1.78
Body mass (kg)	68.7 (11.0)	68 (61–75)	43–110
Follow-up duration (years)	10.2 (1.6)	10.0 (9–11)	5–18

**Table 2 biomedicines-13-02955-t002:** Bone Density Measures and Fracture Risk Per 1 SD or 0.1 g/cm^2^ Decrease.

Parameter	Fracture Outcome	HR (95% CI)	*p* Value	*N Pts*	*N Frax*
Femoral neck BMD	Any fracture	1.43 (1.32–1.54)	<0.001	1214	479
	Hip fracture	2.24 (1.46–3.44)	<0.001	1214	21
	Vertebral fracture	1.44 (1.15–1.81)	0.002	1214	56
Lumbar spine BMD	Any fracture	1.12 (1.04–1.19)	0.001	870	317
	Hip fracture	1.28 (0.92–1.79)	0.149	870	15
	Vertebral fracture	1.33 (1.07–1.65)	0.009	870	37
Femoral neck T-score	Any fracture	1.28 (1.17–1.40)	<0.001	1210	478
	Hip fracture	2.10 (1.28–3.46)	0.003	1210	21
	Vertebral fracture	1.28 (0.98–1.67)	0.072	1210	56
Lumbar spine T-score	Any osteoporotic fracture	1.14 (1.04–1.26)	0.006	1133	438
	Hip fracture	1.30 (0.81–2.09)	0.278	1133	21
	Vertebral fracture	1.53 (1.13–2.08)	0.006	1133	52

**Table 3 biomedicines-13-02955-t003:** Categorical Analysis of Vertebral Fracture Risk Across DXA Parameters.

Parameter	Group	HR (95% CI)	*p* Value
Lumbar spine BMD (g/cm^2^)	≥0.95	Reference	-
	0.85–0.95	1.31 (0.67–2.56)	0.427
	0.75–0.85	1.25 (0.66–2.39)	0.497
	<0.75	2.04 (1.09–3.83)	0.026
Femoral neck BMD (g/cm^2^)	≥0.95	Reference	-
	0.85–0.95	0.86 (0.47–1.57)	0.611
	0.75–0.85	1.38 (0.64–2.96)	0.407
	<0.65	2.40 (1.11–5.19)	0.027
Lumbar spine T-score	>−1	Reference	-
	≤−4	2.96 (1.22–7.18)	0.016

**Table 4 biomedicines-13-02955-t004:** Categorical Analysis of Hip Fracture Risk Across DXA Parameters.

Parameter	Category	HR (95% CI)	*p* Value
Lumbar spine BMD (g/cm^2^)	≥0.95	Reference	-
	0.85–0.95	0.93 (0.27–3.19)	0.907
	0.75–0.85	2.21 (0.74–6.55)	0.154
	<0.75	2.09 (0.82–5.29)	0.121
Femoral neck BMD (g/cm^2^)	≥0.95	Reference	-
	0.85–0.95	0.74 (0.18–3.10)	0.683
	0.75–0.85	2.89 (0.68–12.29)	0.150
	<0.65	6.84 (1.57–29.79)	0.010
Lumbar spine T-score	>−1	Reference	-
	≤−4	1.93 (0.70–5.34)	0.202

**Table 5 biomedicines-13-02955-t005:** Categorical Analysis of All-site Fracture Risk Across DXA Parameters.

Parameter	Category	HR (95% CI)	*p* Value
Lumbar spine BMD (g/cm^2^)	≥0.95	Reference	-
	0.85–0.95	1.03 (0.82–1.28)	0.816
	0.75–0.85	1.03 (0.82–1.29)	0.786
	<0.75	1.45 (1.15–1.82)	0.002
Femoral neck BMD (g/cm^2^)	≥0.95	Reference	-
	0.85–0.95	1.14 (0.92–1.42)	0.237
	0.75–0.85	1.05 (0.81–1.36)	0.718
	<0.65	2.97 (2.24–3.94)	<0.001
Lumbar spine T-score	>−1	Reference	-
	≤−4	1.38 (1.04–1.82)	0.026

**Table 6 biomedicines-13-02955-t006:** Discriminative Performance of DXA Parameters for Vertebral Fractures.

Parameter	AUC (95% CI)	Sensitivity (%)	Specificity (%)	Cut-Off	*N Pts*	*N Frax*
Lumbar spine BMD (g/cm^2^)	0.639 (0.545–0.733)	56.8	66.7	0.86	870	37
Femoral neck BMD (g/cm^2^)	0.596 (0.519–0.672)	64.3	57.7	0.75	1214	56
Lumbar spine T-score	0.630 (0.555–0.706)	67.3	54.9	−2.40	1133	52
Femoral neck T-score	0.597 (0.523–0.671)	53.6	68.7	−2.00	1210	56
Lumbar spine Z-score	0.625 (0.543–0.708)	36.0	85.9	−1.98	1113	50
Femoral neck Z-score	0.593 (0.516–0.671)	51.9	67.4	−0.37	1190	54

**Table 7 biomedicines-13-02955-t007:** Discriminative Performance of DXA Assessments for Hip Fractures.

Parameter	AUC (95% CI)	Sensitivity (%)	Specificity (%)	Optimal Cut-Off	*N Pts*	*N Frax*
Lumbar spine BMD (g/cm^2^)	0.608 (0.434–0.783)	60.0	66.4	0.86	870	15
Femoral neck BMD (g/cm^2^)	0.732 (0.627–0.836)	81.0	61.1	0.74	1214	21
Lumbar spine T-score	0.565 (0.426–0.705)	47.6	69.2	−2.90	1133	21
Femoral neck T-score	0.720 (0.605–0.834)	61.9	75.0	−2.12	1210	21
Lumbar spine Z-score	0.476 (0.325–0.628)	38.1	75.5	0.27	1113	21
Femoral neck Z-score	0.678 (0.556–0.799)	66.7	67.2	−0.38	1190	21

**Table 8 biomedicines-13-02955-t008:** Discriminative Performance of DXA Assessments for All-site Fractures.

Parameter	AUC (95% CI)	Sensitivity (%)	Specificity (%)	Optimal Cut-Off	*N Pts*	*N Frax*
Lumbar spine BMD (g/cm^2^)	0.616 (0.576–0.655)	47.3	73.2	0.86	870	317
Femoral neck BMD (g/cm^2^)	0.621 (0.589–0.653)	53.7	66.1	0.74	1214	479
Lumbar spine T-score	0.602 (0.568–0.636)	50.9	66.2	−2.54	1133	438
Femoral neck T-score	0.619 (0.587–0.651)	63.8	54.4	−1.42	1210	478
Lumbar spine Z-score	0.560 (0.526–0.595)	32.2	78.1	−1.50	1113	428
Femoral neck Z-score	0.583 (0.550–0.615)	85.2	26.6	0.89	1190	467

**Table 9 biomedicines-13-02955-t009:** Relative risk (RR) of fracture according to T-score/BMD in selected epidemiological studies.

Study (Population)	Osteoporotic Fracture (95% CI)	Vertebral Fracture (95% CI)	Femoral Neck Fracture (95% CI)
Current study (*n* = 1.311, Poland)	1.14 (spine, 1.04–1.26); 1.28 (neck, 1.17–1.40)	1.53 (spine, 1.13–2.08); 1.28 (neck, 0.98–1.67)	1.30 (spine, 0.81–2.09); 2.1 (neck, 1.28–3.46)
SOF (*n* = 9.704, USA) [18]	1.33 (spine, 1.2–1.5); 1.44 (neck, 1.3–1.6)	2.06 (spine, 1.9–2.2); 2.01 (neck, 1.8–2.2)	1.49 (spine, 1.3–1.7); 2.20 (neck, 2.0–2.4)
Marshall et al. (~90,000 person-years, meta-analysis) [5]	-	2.30 (spine, 1.9–2.8); 1.80 (neck, 1.6–2.0)	1.60 (spine, 1.4–1.8); 2.60 (neck, 2.0–3.5)
The Manitoba BMD Registry (*n* = 54,459) [19]	1.84 (BMD neck 1.80–1.89)	1.97 (BMD neck 1.91–2.03)	2.99 (BMD neck 2.84–3.15)
DOPS study (*n* = 872, Denmark) [20]	1.30 (spine, 1.1–1.6); 1.32 (neck, 1.1–1.6)	-	-

SOF—Study of Osteoporotic Fracture, DOPS—Danish Osteoporosis Prevention Study, RR—Relative Risk, CI—Confidence Interval.

## Data Availability

The original contributions presented in this study are included in the article. Further inquiries can be directed to the corresponding author.

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
