# Peer review of "What Is the Value of DXA in Predicting Fracture Risk in Postmenopausal Women? A 10-Year Follow-Up Study in the Małopolska Region"

_biomedicines, 2025, doi:10.3390/biomedicines13122955_

Round 1

Reviewer 1 Report (Previous Reviewer 4)

Comments and Suggestions for Authors

The number separator should be a comma, not a dot, to avoid confusion with decimal.

The authors mentioned the limitation in sample size several times. I wonder if a priori sample calculation was performed.

The small number of fractures could lead to instability of the models, as seen by a wide CI, as in the case of lumbar spine T-score (1.22-7.18). The authors mentioned this briefly in the discussion.

The hazard ratios have not been adjusted for potential confounders, I believe. This is a major limitation.

Figure 2 seems redundant when the results are already presented in Table 2, unless the authors do a subanalysis based on potential confounders.

Author Response

Thank you very much for taking the time to review this manuscript. Please find the detailed responses below and the corresponding revisions in the re-submitted file. 

Point-by-point response to Comments and Suggestions for Authors:  

Comments 1: The number separator should be a comma, not a dot, to avoid confusion with decimal.
Response 1: We have followed the numeric style used in previously published Biomedicines articles, where dots were applied as thousand separators.

Comments 2: The authors mentioned the limitation in sample size several times. I wonder if a priori sample calculation was performed.
Response 2: Thank you for your insightful comment. We agree that the sample size is an important limitation of our study. As the study was part of a larger research project, an a priori sample size calculation was not performed at the design stage. 

Comments 3: The small number of fractures could lead to instability of the models, as seen by a wide CI, as in the case of lumbar spine T-score (1.22-7.18). The authors mentioned this briefly in the discussion.
Response 3: Comments have been added to the discussion. 

Comments 4: The hazard ratios have not been adjusted for potential confounders, I believe. This is a major limitation.
Response 4: Comments have been added to the discussion. 

Comments 5: Figure 2 seems redundant when the results are already presented in Table 2, unless the authors do a subanalysis based on potential confounders.
Response 5: Figure 2 and all references to it have been removed.

Reviewer 2 Report (New Reviewer)

Comments and Suggestions for Authors

I carefully reviewed the present manuscript that investigates the long-term predictive value of DXA-based bone mineral density (BMD) in postmenopausal women from the Małopolska region. The study is interesting, as population-specific data on fracture risk prediction remain limited. However, despite the large cohort and extended follow-up, several methodological and interpretative issues must be addressed before publication.

Please accept the following criticisms:

  • The introduction is overly descriptive and includes epidemiological data already well established in the literature. It should be shortened and focused on the knowledge gap—namely, the limited availability of prospective data from Polish populations—and the specific rationale for this study.
  • The results largely replicate existing data from other European cohorts. The authors should clearly state what is the novelty brought by their study—e.g., whether ethnic, regional, or methodological differences justify this investigation.
  • The study design is described as “prospective,” yet data collection partly relies on retrospective review and self-reported fractures after telephone follow-up. The authors should clearly define the design (prospective vs. retrospective cohort) and discuss possible recall bias.
  • The definition and diagnostic criteria for fractures are unclear, which may introduce significant bias into the study.
  • Only 34% of participants underwent active radiologic screening; in others, fractures were based on patient report or medical documentation. This introduces underestimation bias. The authors should explicitly acknowledge this limitation and provide data on interobserver validation or diagnostic confirmation rates.
  • The analysis does not account for key modifiers such as medication use, hormonal therapy, or comorbidities (e.g., rheumatoid arthritis, diabetes). Adjusted models including these variables would strengthen the conclusions.
  • The Cox regression and ROC analyses are well presented but lack clarity regarding the adjustment variables used and the handling of missing data. The term “nonlinear risk” should be supported with a spline or categorical model visualization (e.g., restricted cubic spline for T-score vs. fracture risk).
  • The Discussion reiterates numerical results without adequate clinical interpretation. The authors should discuss why the association between BMD and fracture risk is nonlinear and what implications this has for clinical decision-making or FRAX thresholds.
  • The conclusion should better emphasize how DXA-derived thresholds may inform treatment stratification in Polish women and whether new cut-offs (different from WHO) are proposed.
  • The English is generally adequate but verbose; certain expressions (e.g., “represents the most serious complication”) should be replaced with concise scientific phrasing. Minor grammatical and stylistic revision is recommended.
  • The Methods section should state the ethical approval reference and whether informed consent covered long-term data reuse.
  • At the end of the introduction, there is a sentence that appears crossed out, as if track changes were still active. Please correct this.
  • Units should be consistent (e.g., g/cm² vs. g/cm2).

Author Response

Thank you very much for taking the time to review this manuscript. Please find the detailed responses below and the corresponding revisions in the re-submitted file.
Point-by-point response to Comments and Suggestions for Authors:  

Comments 1: The introduction is overly descriptive and includes epidemiological data already well established in the literature. It should be shortened and focused on the knowledge gap—namely, the limited availability of prospective data from Polish populations—and the specific rationale for this study.

Response 1: The paragraph containing publicly available epidemiological information has been removed. The reviewer's suggestions regarding limited availability and the rationale for the study have been added. 

Comments 2: The results largely replicate existing data from other European cohorts. The authors should clearly state what is the novelty brought by their study—e.g., whether ethnic, regional, or methodological differences justify this investigation.
Response 2: The introduction has been modified in accordance with these comments.

Comments 3: The study design is described as “prospective,” yet data collection partly relies on retrospective review and self-reported fractures after telephone follow-up. The authors should clearly define the design (prospective vs. retrospective cohort) and discuss possible recall bias.
Response 3: Some of the data was collected retrospectively, based on documentation, and then the patients were observed- this was clearly described in the methodology.

Comments 4: The definition and diagnostic criteria for fractures are unclear, which may introduce significant bias into the study.
Response 4: The definitions and fracture criteria were clear to the trained investigators and were based, among others, on the Genant criteria for vertebral compression fractures; however, this was not described in the manuscript. The main difficulty was obtaining all source imaging data (X-ray and CT results) for verification, which has been addressed in the Discussion in the context of the low number of identified fractures. 

Comments 5: Only 34% of participants underwent active radiologic screening; in others, fractures were based on patient report or medical documentation. This introduces underestimation bias. The authors should explicitly acknowledge this limitation and provide data on interobserver validation or diagnostic confirmation rates.
Response 5: This point has been addressed and incorporated into the revised Discussion.

Comments 6:The analysis does not account for key modifiers such as medication use, hormonal therapy, or comorbidities (e.g., rheumatoid arthritis, diabetes). Adjusted models including these variables would strengthen the conclusions.
Response 6: Thank you for valuable comment. We fully agree with your observation. However, we were unable to perform this analysis due to the lack of sufficient data. 

Comments 7: The Cox regression and ROC analyses are well presented but lack clarity regarding the adjustment variables used and the handling of missing data. The term “nonlinear risk” should be supported with a spline or categorical model visualization (e.g., restricted cubic spline for T-score vs. fracture risk).
Response 7: Due to variable event rates across fracture outcomes (ranging from EPV=15 to EPV=479), power and convergence concerns and consistency in model specification, we present unadjusted Cox regression analyses as the main analyses. For selected outcomes, we present additional age-adjusted sensitivity analyses in the Appendix section. Our analyses remain relatively robust to the effects of age (about 3-9% change in HRs - reduced magnitude), which was identified as the main biological confounder. The highly variable EPV rate and associated power concerns limits the reliability of the age, and age+BMI adjusted model for all different outcome types. We performed exploratory sensitivity analyses regarding the effects of BMI, which suggest opposite confounding directions with respect to age. To allow for comparison across all fracture types, we present crude models as primary analyses, and age-adjusted analyses in Appendix, though upon the Reviewer's request, we may add further exploratory analyses in the Appendix section. 

Comments 8: The Discussion reiterates numerical results without adequate clinical interpretation. The authors should discuss why the association between BMD and fracture risk is nonlinear and what implications this has for clinical decision-making or FRAX thresholds.
Response 8: The conclusions have been edited as suggested.

Comments 9: The conclusion should better emphasize how DXA-derived thresholds may inform treatment stratification in Polish women and whether new cut-offs (different from WHO) are proposed.
Response 9: The conclusions have been edited as suggested.

Comments 10: The English is generally adequate but verbose; certain expressions (e.g., “represents the most serious complication”) should be replaced with concise scientific phrasing. Minor grammatical and stylistic revision is recommended.
Response 10: The introduction has been changed as suggested.

Comments 11: The Methods section should state the ethical approval reference and whether informed consent covered long-term data reuse.
Response 11: The EC approval number has been added to the Methodology section. At the time of the study, there were no regulations in Poland concerning the long-term use of data.

Comments 12: At the end of the introduction, there is a sentence that appears crossed out, as if track changes were still active. Please correct this.
Response 12: We thank the reviewer for noticing this. The formatting error has been corrected. 

Comments 13: Units should be consistent (e.g., g/cm² vs. g/cm2).
Response 13: All units have been carefully reviewed and standardized. 

Reviewer 3 Report (New Reviewer)

Comments and Suggestions for Authors

Manuscript Title: What is the value of DXA in predicting fracture risk in postmenopausal women? A 10-year follow-up study in the Małopolska region
Recommendation: Minor Revision

Summary

This study reports a 10-year follow-up analysis of 1311 postmenopausal women in the Małopolska region of Poland, evaluating the predictive value of DXA-derived BMD and T-scores for osteoporotic fracture risk. The dataset and statistical approaches (Cox regression, ROC analysis) are appropriate, and the results align with large international cohorts (e.g., SOF, Marshall et al., BMJ 1996). Overall, the paper is well-structured and the conclusions are clear.However, several methodological and interpretive issues require clarification or expansion, primarily within the Discussion section. These revisions are interpretive and do not require new experiments.

Major Comments

  1. Novelty and Scientific Significance (L17–35, L188–237)
    The study design is solid but mainly reproduces previously known findings. The authors should more clearly state the study’s unique contribution, such as:
  • The value of validating DXA-based fracture prediction specifically in the Polish population;
  • How lifestyle, ethnicity, or healthcare access may differentiate this cohort from prior international studies;
  • The potential public-health implications for regional osteoporosis management or screening policies.
  1. Limited Covariate Adjustment in Cox Models (L99–111; see also Abstract L25)
    Age appears to be the only covariate included. Important confounders—BMI, prior fractures, smoking, medication use, menopausal duration—were not reported or adjusted.
    Please clarify whether multivariable adjustment was tested. If not, acknowledge this in the Discussion and discuss how unmeasured confounding may affect the hazard ratios.
  2. ROC Interpretation and Figure Labeling (L157–187, L163–164)
  • Several AUCs < 0.70 (Tables 6–8) are described as “moderate discrimination,” which overstates the predictive performance.
  • Figure 3 is mislabeled as a “Forest plot.”
    Please:
  1. Rephrase AUC < 0.70 as “limited” or “modest discrimination”;
  2. Correct the figure title and legend;
  3. If feasible, note whether AUC differences between measurement sites were statistically significant (e.g., DeLong test) or state that such analysis was not performed.
  1. Small Number of Events and Wide Confidence Intervals (L114–117; L147–152; Table 4)
    Only 22 hip-fracture events were observed, yielding very wide CIs (e.g., HR 6.84 [1.57–29.79]). This reduces model stability.
    Please emphasize in the Discussion that these estimates should be interpreted cautiously and validated in larger datasets.
  2. Lack of Adjustment for Osteoporosis Treatment (L211–213)
    Roughly 17 % of participants received therapy during follow-up, yet treatment exposure was not modeled.
    Please add a statement acknowledging that treatment may have modified fracture incidence and DXA predictive performance.
  3. Outdated References in the Introduction (L37–58)
    The introduction cites mostly older sources from the 1990s.
    Please update the background with recent reviews or meta-analyses (e.g., J Bone Miner Res 2023; 38: 1850–1862) to reflect the current state of knowledge.

Language and Reference Issues

  1. Incorrect Year in Reference List (L345–346)
    The DOPS study is listed as “J Bone Miner Res. 2026; 21: 796–800.” The year should be 2006.
    Please correct this and check the entire reference list for similar typographical or formatting inconsistencies.

Overall Recommendation

Decision: Minor Revision

The manuscript offers valuable long-term regional data confirming the predictive utility of DXA but requires stronger discussion of model limitations, treatment effects, and updated literature.
No new experiments are necessary; revisions should focus on textual and interpretive improvements.

Author Response

Thank you very much for taking the time to review this manuscript. Please find the detailed responses below and the corresponding revisions in the re-submitted file. 

Point-by-point response to Comments and Suggestions for Authors:  
Comments 1: Novelty and Scientific Significance (L17–35, L188–237)
The study design is solid but mainly reproduces previously known findings. The authors should more clearly state the study’s unique contribution, such as: The value of validating DXA-based fracture prediction specifically in the Polish population; How lifestyle, ethnicity, or healthcare access may differentiate this cohort from prior international studies; The potential public-health implications for regional osteoporosis management or screening policies.
Response 1: Thank you for helpful comment. These points have been incorporated into the revised versions of the Introduction and Discussion sections. 

Comments 2:Limited Covariate Adjustment in Cox Models (L99–111; see also Abstract L25)
Age appears to be the only covariate included. Important confounders—BMI, prior fractures, smoking, medication use, menopausal duration—were not reported or adjusted. Please clarify whether multivariable adjustment was tested. If not, acknowledge this in the Discussion and discuss how unmeasured confounding may affect the hazard ratios.
Response 2: Due to variable event rates across fracture outcomes (ranging from EPV=15 to EPV=479), power and convergence concerns and consistency in model specification, we present unadjusted Cox regression analyses as the main analyses. For selected outcomes, we present additional age-adjusted sensitivity analyses in the Appendix section. Our analyses remain relatively robust to the effects of age (about 3-9% change in HRs - reduced magnitude), which was identified as the main biological confounder. The highly variable EPV rate and associated power concerns limits the reliability of the age, and age+BMI adjusted model for all different outcome types. We performed exploratory sensitivity analyses regarding the effects of BMI, which suggest opposite confounding directions with respect to age. To allow for comparison across all fracture types, we present crude models as primary analyses, and age-adjusted analyses in Appendix, though upon the Reviewer's request, we may add further exploratory analyses in the Appendix section. 

Comments 3: ROC Interpretation and Figure Labeling (L157–187, L163–164)
Several AUCs < 0.70 (Tables 6–8) are described as “moderate discrimination,” which overstates the predictive performance. Figure 3 is mislabeled as a “Forest plot.” Please: 1.Rephrase AUC < 0.70 as “limited” or “modest discrimination”; 2.Correct the figure title and legend; 3.If feasible, note whether AUC differences between measurement sites were statistically significant (e.g., DeLong test) or state that such analysis was not performed.
Response 3:All comments have been addressed and corrected.

Comments 4: Small Number of Events and Wide Confidence Intervals (L114–117; L147–152; Table 4) Only 22 hip-fracture events were observed, yielding very wide CIs (e.g., HR 6.84 [1.57–29.79]). This reduces model stability. Please emphasize in the Discussion that these estimates should be interpreted cautiously and validated in larger datasets.
Response 4: The discussion was amended to take these comments into account.

Comments 5: Lack of Adjustment for Osteoporosis Treatment (L211–213)
Roughly 17 % of participants received therapy during follow-up, yet treatment exposure was not modeled. Please add a statement acknowledging that treatment may have modified fracture incidence and DXA predictive performance.
Response 5: Thank you for your valuable comment. The Discussion section has been revised to incorporate these suggestions.

Comments 6: Outdated References in the Introduction (L37–58)
The introduction cites mostly older sources from the 1990s. Please update the background with recent reviews or meta-analyses (e.g., J Bone Miner Res 2023; 38: 1850–1862) to reflect the current state of knowledge.
Response 6: Thank you for this comment. The introduction contains only one reference from the 1990s, which is included because it remains a key work for this specific topic. In response to the suggestion, we have also added two more recent citations to further strengthen and update the introductory section.

Comments 7: Incorrect Year in Reference List (L345–346)
The DOPS study is listed as “J Bone Miner Res. 2026; 21: 796–800.” The year should be 2006.
Please correct this and check the entire reference list for similar typographical or formatting inconsistencies.
Response 7: We thank the reviewer for noting this error. The publication year of the DOPS study has been corrected to 2006, and the entire reference list has been thoroughly reviewed for typographical, formatting, and citation inconsistencies. Minor corrections were applied where necessary.

This manuscript is a resubmission of an earlier submission. The following is a list of the peer review reports and author responses from that submission.

Round 1

Reviewer 1 Report

Comments and Suggestions for Authors

The paper is actually well-written but it has a limited novelty, since, if I understood correctly, it essentially confirms in Polish women results previously obtained in other populations of Caucasian women. An interesting aspect is the non-linear relationship between fracture risk and T-score, but the Authors should better comment on the fact that for T-score<-3 the risk of femoral neck fracture decreases.

Further concerns are the following:

  1. Why those data have submitted just now if the follow-up was completed about 12 years ago? The Authors should provide a reasonable justification.
  2. The Authors do not report any study limitations and they are encouraged to add and explain them.

Author Response

Thank you very much for taking the time to review this manuscript. Please find the detailed responses below and the corresponding revisions in the re-submitted file. 

Point-by-point response to Comments and Suggestions for Authors:  
Comments 1: The paper is actually well-written but it has a limited novelty, since, if I understood correctly, it essentially confirms in Polish women results previously obtained in other populations of Caucasian women.An interesting aspect is the non-linear relationship between fracture risk and T-score, but the Authors should better comment on the fact that for T-score<-3 the risk of femoral neck fracture decreases.
Response 1: We are grateful for this valuable comment. We agree that our study mainly confirms previous findings, but we believe it provides important population-specific data for Polish women, where such evidence is scarce. The apparent decrease in femoral neck fracture risk at T-scores <−3 is most likely due to the small sample size in this subgroup and the resulting limited statistical power. These may have been individuals with additional factors that provided protection, but this is merely speculation since we cannot verify it based on the available source data. Interestingly, a hip T-score < -3.0 was associated with the highest risk of vertebral compression fractures. In this group, we had 62 compression fractures, nearly 100% more than hip fractures. The small number of femoral neck fractures (37) may also be another factor influencing our statistics in this way.

Comments 2: Why those data have submitted just now if the follow-up was completed about 12 years ago? The Authors should provide a reasonable justification.
Response 2: The study was originally conducted as part of a doctoral project, which was defended in 2017. The delay was due to personal matters of the main author, change of workplace, new specialization, and different field of study. However, after reviewing the literature and verifying that epidemiological data from Poland remain very scarce, we decided that it was worthwhile to publish our data.

Comments 3: The Authors do not report any study limitations and they are encouraged to add and explain them.
Response 3: We thank the Reviewer for this suggestion. The limitations of our study have now been added and explained in the Discussion section.

Reviewer 2 Report

Comments and Suggestions for Authors

1. The study’s conclusions do not differ from those of the current mainstream literature, and therefore the work lacks sufficient novelty.

2. The study includes only participants from a single region; this limited sample restricts the generalizability of the findings and diminishes the overall significance of the work.

Author Response

Thank you very much for taking the time to review this manuscript. Please find the detailed responses below and the corresponding revisions in the re-submitted file. 

Point-by-point response to Comments and Suggestions for Authors:
Comments 1: The study’s conclusions do not differ from those of the current mainstream literature, and therefore the work lacks sufficient novelty.
Response 1: We thank the Reviewer for this observation. We agree that our conclusions are in line with the current mainstream literature. However, we believe that confirming these associations in a large, well-characterized cohort of Polish postmenopausal women is of value, as such population-specific data from Poland remain scarce. The emphasis on the nonlinear increase in fracture risk with decreasing DXA values is meant to highlight the varying utility of DXA in risk stratification, consistent with the concept of very high, high, and low fracture risk. In clinical practice, the simple classification based on a T-score of -2.5 still predominates, which is a personal observation of the authors based on discussions during medical trainings and conferences. This work aims to provide Polish epidemiological data that will assist relevant institutions in optimizing healthcare for osteoporosis

Comments 2: The study includes only participants from a single region; this limited sample restricts the generalizability of the findings and diminishes the overall significance of the work.
Response 2: We agree that focusing on a single region limits the generalizability of our results. Still, this is one of the few long-term studies in a Polish female cohort, and the findings are consistent with international data, which supports their broader relevance. Secondly, Poland is a homogeneous country where almost 100% of the population aged 50+ are Caucasian women, not Hispanic. This allows for a high probability of extrapolating the data to the entire Polish population aged 50+.

Reviewer 3 Report

Comments and Suggestions for Authors

NHANES III never included lumbar-spine BMD for the T-score reference. ISCD instructs: NHANES III reference data for hip; manufacturer database for spine.

If analysis was performed for T-score intervals lower than -1.0, why not analyse intervals of higher values too?

Are there any new data available instead of the reference from Marshall et al. (1996)? Why?

There is a reason why the T-score is used instead of BMD; please take this into account and discuss accordingly. 

There was a multiple use of the acronym DXA with a vague meaning. For instance: 'Our findings confirmed that DXA at the femoral neck correlates most strongly with  
hip fracture risk.' -What is DXA: BMD, T-score, Z-score, or all of them together?

I understand that there was a wish to repeat older analysis in the new cohort, but it should be explained why this analysis is important and what new findings it could provide.  

References 5 and 16 are the same.

Reference 6 does not fit the text. 

Correct reference 7.

Comments on the Quality of English Language

'A decrease in hip BMD and hip T-score' suggests a relative change of BMD between measurements; the word low should be used. 

Please use 'vertebral fracture' instead of 'compression fracture'.

Author Response

Thank you very much for taking the time to review this manuscript. Please find the detailed responses below and the corresponding revisions in the re-submitted file. 

Point-by-point response to Comments and Suggestions for Authors:
Comments 1: NHANES III never included lumbar-spine BMD for the T-score reference. ISCD instructs: NHANES III reference data for hip; manufacturer database for spine.
Response 1: We thank you for this comment. We agree that NHANES III provides reference data only for the hip, while lumbar spine T-scores are derived from the manufacturer’s database, in line with ISCD recommendations. In our study, lumbar spine values were based on the Hologic reference database. This has now been clarified in the Materials and Methods section of the revised manuscript.

Comments 2: If analysis was performed for T-score intervals lower than -1.0, why not analyse intervals of higher values too?
Response 2: Our primary aim was to investigate the predictive value of low BMD in the osteopenic and osteoporotic range, where the risk of fractures is clinically most relevant. Therefore, the interval analysis was focused on T-score values below −1.0. The topic is very interesting, but further analyses would exceed the aim of the article. Additional results would require extra tables, conclusions, and discussion, which would dilute the main purpose of the work, which was to compare the Polish population to the reference studies. The T-score intervals used in those studies were the ones we adopted. Additionally, we were constrained by editorial limits regarding the length and scope of the article. In the subgroup with T-scores higher than −1.0 (normal BMD), the number of fractures was very small, which limited the statistical power and did not allow for meaningful comparisons. 

Comments 3: Are there any new data available instead of the reference from Marshall et al. (1996)? Why?Response 3: The study by Marshall et al. (1996) was cited as it represents the first landmark meta-analysis establishing the relationship between BMD and fracture risk, and it has been referenced in most subsequent guidelines. We chose the Marshall study even though it dates back to 1996. Of course, there are many data confirming these findings, but the Marshall study was groundbreaking. It is also very well known, and the conclusions from this work are widely accepted. I understand the concerns related to using a study from over 20 years ago, but its value is timeless

Comments 4: There is a reason why the T-score is used instead of BMD; please take this into account and discuss accordingly. 
Response 4: We analyzed both parameters. In each table, values for BMD of the spine and hip are provided and can be compared with the T-score. However, only T-scores were chosen for the figures. The BMD and T-score data are similar, and these images would overlap. The figures would be unreadable. We consistently chose the T-score because physicians better understand the T-score indicator and primarily use it in practice.

Comments 5: There was a multiple use of the acronym DXA with a vague meaning. For instance: 'Our findings confirmed that DXA at the femoral neck correlates most strongly with 
hip fracture risk.' -What is DXA: BMD, T-score, Z-score, or all of them together?
Response 5: In this case, the emphasis was to highlight that all analyzed hip DXA parameters (hip BMD, T-score, and Z-score) best predict hip fractures. The sentence in the discussion has been changed and corrected accordingly

Comments 6: I understand that there was a wish to repeat older analysis in the new cohort, but it should be explained why this analysis is important and what new findings it could provide.  
Response 6: This comment has been addressed in the new version of the discussion.

Comments 7: References 5 and 16 are the same. Reference 6 does not fit the text. Correct reference 7. 
Response 7: Thank you for your vigilance. Citation 16 has been removed and replaced with 5. Citation 6 has been modified, and 7 corrected.

Response to Comments on the Quality of English Language:
Point 1: 'A decrease in hip BMD and hip T-score' suggests a relative change of BMD between measurements; the word low should be used. 
Response 1:  The phrase “a decrease in hip BMD and hip T-score” has been replaced with “low hip BMD and low hip T-score” in the relevant sentence, as suggested.

Point 2: Please use 'vertebral fracture' instead of 'compression fracture'.
Response 2:  We appreciate your comment. The term “compression fracture” has been replaced with “vertebral fracture” throughout the manuscript to ensure consistency with standard terminology.

The language and graphic tables have already been partially corrected according to precise suggestions. As for the graphics, please provide more precise comments if there are still any objections.

Reviewer 4 Report

Comments and Suggestions for Authors

Throughout: When writing the numbers, use a comma as the separator, not dots. Examples: 1,284, not 1.284.

Title: No need to mention the number of subjects in the title.

The methods contain only limited details about the study. Further elaboration will be needed. Specifically,

Please clarify the DXA reference… reference from NHANES III Caucasian database

Reason for lost to follow up should be indicated.

What questionnaire was used at the initial phase? What questions were asked during the follow up?

Were the 3350 subjects screened using the same DXA?

Medical records were derived from which hospital? Are all the subjects from the same hospital?

Fracture determination: Is it solely based on the patient’s self-declaration?

Statistical test: What test did the authors use to assess normality.

Results:

Please replace p-values=0.000 to p < 0.001 throughout.

For ROC analysis, I will be interested to know the optimal cutoff BMD values to predict fractures. In addition, the sensitivity and specificity for the cutoff values should be indicated.

Discussion: Table 9: Consider expanding the range of the studies included, cite the studies and provided  the definitions of abbreviations as footnotes. The RR values cited should be accompanied by 95% CI values. The exclusion of other risk factors of fracture should also be mentioned.

Limitations of the study should be mentioned. Importantly, if fracture determination is based on patients’ self-declaration, it might not be accurate.

Comments on the Quality of English Language

Some level of proofreading will be necessary. 

Author Response

Thank you very much for taking the time to review this manuscript. Please find the detailed responses below and the corresponding revisions in the re-submitted file. 

Point-by-point response to Comments and Suggestions for Authors:
Comments 1: Throughout: When writing the numbers, use a comma as the separator, not dots. Examples: 1,284, not 1.284.
Response 1: We have followed the numeric style used in previously published Biomedicines articles, where dots were applied as thousand separators. The problem will be reported to the editorial team and will be subject to their decision.

Comments 2: Title: No need to mention the number of subjects in the title.
Response 2: We thank you for this helpful suggestion. In accordance, we have modified the title by removing the number of subjects.

Comments 3: The methods contain only limited details about the study. Further elaboration will be needed. Specifically, Please clarify the DXA reference… reference from NHANES III Caucasian database. Reason for lost to follow up should be indicated. What questionnaire was used at the initial phase? What questions were asked during the follow up? Were the 3350 subjects screened using the same DXA? Medical records were derived from which hospital? Are all the subjects from the same hospital?
Response 3: Materials and Methods section has been modified according to the above suggestions.

Comments 4: Fracture determination: Is it solely based on the patient’s self-declaration?
Response 4: : Materials and Methods section has been modified according to the above suggestions.

Comments 5: Statistical test: What test did the authors use to assess normality.
Response 5: We used the chi-square test of independence to analyze associations between categorical variables. When expected frequencies in any cell of the contingency table were below 5, Fisher’s exact test was applied for 2x2 tables, and Yates’s correction was used for the chi-square test in other cases. Quantitative variables with normal distribution were described by mean and standard deviation, otherwise by median and quartiles. The Shapiro-Wilk test was used to assess normality.

Comments 6: Results: Please replace p-values=0.000 to p < 0.001 throughout. For ROC analysis, I will be interested to know the optimal cutoff BMD values to predict fractures. In addition, the sensitivity and specificity for the cutoff values should be indicated.
Response 6: The p-values = 0.000 have been replaced with p < 0.001. We searched for the minimum lumbar spine BMD value above which compression fractures do not occur. Similarly, we analysed the minimum hip BMD value above which hip fractures do not occur. The results show that there is no minimum "safe" lumbar spine BMD value above which fractures are absent. Vertebral fractures occur even at BMD values greater than 1.0 g/cm². For the hip, a minimum BMD value can be suggested because fractures start occurring at BMD below 0.97 g/cm², but in practice, these values are very rare.

Comments 7: Discussion: Table 9: Consider expanding the range of the studies included, cite the studies and provided  the definitions of abbreviations as footnotes. The RR values cited should be accompanied by 95% CI values. The exclusion of other risk factors of fracture should also be mentioned.
Response 7: We have revised Table 9 according to the suggestions. 

Comments 8: Limitations of the study should be mentioned. Importantly, if fracture determination is based on patients’ self-declaration, it might not be accurate.
Response 8: We thank the Reviewer for this suggestion. The limitations of our study have now been added and explained in the Discussion section. Fractures were assessed based on interviews, medical documentation (discharge summaries and imaging results), which has been updated in section 2 (Materials and Methods).

The language and graphic tables have already been partially corrected according to precise suggestions. As for the graphics, please provide more precise comments if there are still any objections.

Reviewer 5 Report

Comments and Suggestions for Authors

This study aims to evaluate the predictive value of BMD measurements for vertebral, hip, and other low-energy fractures in women aged 50 years and older. It presents an analysis based on a large cohort with long-term follow-up in a Polish population of postmenopausal Caucasian women.

General Comments

Population characteristics: Please provide more details about the characteristics of the studied population, with particular attention to risk factors for low bone density.

Treatment during follow-up: Clarify which treatments were received by patients during the follow-up period, as these could influence fracture risk outcomes.

Vertebral fracture assessment: Specify the method used for vertebral fracture assessment (clinical evaluation, VFA, or radiography) since that could affect considerably the fracture frequency because in this condition vertabral fracture are often asymptomatic.

Tables 6, 7, and 8: Revise the titles to be more descriptive and self-explanatory, so they can be understood independently of the figures.

Table 9: Consider reorganizing the layout by switching columns to rows for better readability. Please also add references for the studies cited and indicate the study population at the end of each category.

Patient distribution in tables: For all tables presenting categories of patients, include the number of patients in each category to improve clarity and interpretation.

Line 202 : The results discussed here should be analyzed in relation to different population parameters (e.g., number of patients, gender, age, fracture risk factors and importantly with osteoporosis treatment received). This is essential to explain why fracture risk did not increase with decreasing BMD, a finding that must be interpreted with great caution before drawing firm conclusions.

Author Response

Thank you very much for taking the time to review this manuscript. Please find the detailed responses below and the corresponding revisions in the re-submitted file. 

Point-by-point response to Comments and Suggestions for Authors:
Comments 1: Population characteristics: Please provide more details about the characteristics of the studied population, with particular attention to risk factors for low bone density.
Response 1: Materials and Methods section has been modified according to the above suggestions.

Comments 2: Treatment during follow-up: Clarify which treatments were received by patients during the follow-up period, as these could influence fracture risk outcomes.
Response 2: The question is very important. Treatment was evaluated in the study. It was defined as the use of one of the active drugs for at least 12 months (bisphosphonates, strontium ranelate, calcitonin, raloxifene, teriparatide). Denosumab was not available in Poland. This limitation of the study was commented on in the discussion. 

Comments 3: Vertebral fracture assessment: Specify the method used for vertebral fracture assessment (clinical evaluation, VFA, or radiography) since that could affect considerably the fracture frequency because in this condition vertabral fracture are often asymptomatic.
Response 3: Vertebral fractures were assessed in two ways. In patients regularly followed at the osteoporosis clinic, spine X-rays and vertebral fracture assessment (VFA) were performed in cases of back pain, height loss greater than 4 cm, body deformity, and a positive 'fish vertebra' sign on physical examination. In patients who did not receive regular treatment, compression fractures were confirmed based on provided medical documentation. The discussion was updated accordingly.

Comments 4: Tables 6, 7, and 8: Revise the titles to be more descriptive and self-explanatory, so they can be understood independently of the figures.
Response 4: We thank you for this helpful suggestion. In the revised manuscript, we have modified the titles of Tables 6, 7 and 8 to make them more descriptive and self-explanatory, so that each table can be understood independently of the figures.

Comments 5: Table 9: Consider reorganizing the layout by switching columns to rows for better readability. Please also add references for the studies cited and indicate the study population at the end of each category.
Response 5: We have revised Table 9 by reorganizing the layout (switching columns to rows) to improve readability. In addition, we have added the references for each cited study and indicated the study population at the end of each category. 

Comments 6: Patient distribution in tables: For all tables presenting categories of patients, include the number of patients in each category to improve clarity and interpretation.
Response 6: Tables 3, 4, and 5 already have 5 columns, and adding another one containing the number of patients would make the table unreadable. This would hinder the interpretation of the main parameter (RR).

Comments 7: Line 202 : The results discussed here should be analyzed in relation to different population parameters (e.g., number of patients, gender, age, fracture risk factors and importantly with osteoporosis treatment received). This is essential to explain why fracture risk did not increase with decreasing BMD, a finding that must be interpreted with great caution before drawing firm conclusions.
Response 7: The discussion was updated accordingly.

The language and graphic tables have already been partially corrected according to precise suggestions. As for the graphics, please provide more precise comments if there are still any objections.

Round 2

Reviewer 2 Report

Comments and Suggestions for Authors

I acknowledge the author's revised manuscript and agree to publish it.

Author Response

We sincerely thank you for the positive evaluation of our revised manuscript and for supporting its publication. We greatly appreciate your time and constructive comments provided during the review process, which have significantly helped us improve the quality and clarity of the paper.

Reviewer 3 Report

Comments and Suggestions for Authors

The authors addressed some of my comments and improved the manuscript, whose aims and results are now more clearly presented. 

Regarding my comment 3, I suggest discussing at least the Manitoba BMD Registry in light of your results, particularly in the context of population characteristics, as a potential limitation of DXA measurements. Table 9 should be enlarged with these data. 

I would suggest revisiting comment 4 and incorporating a general explanation of the differences between various DXA parameters and their clinical significance into the introduction section.  

I would suggest revising the final detail and omitting the GARVAN calculator from the last sentence, as it is less validated globally. The word "other" might be used. 

Please translate the reference 19. 

Author Response

We sincerely thank you for your careful re-review of our manuscript and for providing additional constructive suggestions. Below we provide detailed responses to each of your remarks and we also attach the revised version of the manuscript with the suggested changes incorporated. 

Comments 1: Regarding my comment 3, I suggest discussing at least the Manitoba BMD Registry in light of your results, particularly in the context of population characteristics, as a potential limitation of DXA measurements. Table 9 should be enlarged with these data. 
Response 1: Thanky. The Manitoba BMD register is  informative. The Discussion has been revised in accordance with your comments. Table 9 has been upgraded. 

Comments 2: I would suggest revisiting comment 4 and incorporating a general explanation of the differences between various DXA parameters and their clinical significance into the introduction section.  
Response 2: The introduction has been revised in accordance with your comment. We analyzed BMD, T-scores, and Z-scores in each patient; however, the results in this study are presented primarily in terms of T-scores. Although BMD represents the primary measurement, the T-score is most commonly used in clinical practice and is the parameter best understood by practicing physicians. Moreover, the registries we relied upon for comparison also reported their results using T-scores.

Comments 3: I would suggest revising the final detail and omitting the GARVAN calculator from the last sentence, as it is less validated globally. The word "other" might be used. 
Response 3: Thank you again for this helpful remark. We have revised the final sentence accordingly by omitting the GARVAN calculator and replacing it with “other” fracture risk calculators.

Comments 4: Please translate the reference 19. 
Response 4: Thank you for pointing this out. We have translated reference 19 into English.

Reviewer 4 Report

Comments and Suggestions for Authors

Table 2, modify to include 95% confidence intervals and the exact p-values (not in the footnotes).

Table 3-5 includes the sample size for each subgroup so that the readers are aware that the subgroups are not too small. Subgroups at the extremes seem to have a large 95% CI range, indicating an unstable relationship, probably due to the very small sample size of that group.

Comment on the clinical usefulness of AUC values of 0.7 or below for each BMD indicator of fracture.

Author Response

We sincerely thank you for your careful re-review of our manuscript and for providing additional constructive suggestions. Below we provide detailed responses to each of your remarks and we also attach the revised version of the manuscript with the suggested changes incorporated. 

Comments 1: Table 2, modify to include 95% confidence intervals and the exact p-values (not in the footnotes).
Response 1: We thank you for this valuable suggestion. Table 2 has been revised. We decided to revise part of the statistical analyses and repeat them using more up-to-date methods. A statistician from a reputable academic center was invited to collaborate, which resulted in the inclusion of additional tables and figures in the article to clearly illustrate certain relationship. The paper includes additional tables with visualizations of the results, which appear to be easier to interpret and more comprehensible for the practicing physician.

Comments 2: Table 3-5 includes the sample size for each subgroup so that the readers are aware that the subgroups are not too small. Subgroups at the extremes seem to have a large 95% CI range, indicating an unstable relationship, probably due to the very small sample size of that group.
Response 2: Tables 3–5 have been modified in response to your comment as part of a repeated analysis using the latest statistical tools, which are better suited to our dataset.

Comments 3: Comment on the clinical usefulness of AUC values of 0.7 or below for each BMD indicator of fracture. Response 3: Thank you for highlighting this point. We agree that AUC values of 0.7 or below indicate only moderate discriminative ability. The corresponding comment has been included in the Discussion section.

Round 3

Reviewer 3 Report

Comments and Suggestions for Authors

"Although BMD represents the primary measurement, the T-score is most commonly used in clinical practice and is the parameter best understood by practicing physicians." - This sentence is factually incorrect.